# The Effect of Pollen on Coral Health

**DOI:** 10.3390/biology12121469

**Published:** 2023-11-27

**Authors:** Triona Barker, Mark Bulling, Vincent Thomas, Michael Sweet

**Affiliations:** 1Aquatic Research Facility, Nature-Based Solutions Research Centre, University of Derby, Derby DE22 1GB, UK; 2Coral Spawning Lab, Unit 6 Midas Metro Centre, 193 Garth Road, Morden SM4 4NE, UK

**Keywords:** pollen, coral mortality, immune response, hay fever, nutrient cascades

## Abstract

**Simple Summary:**

Corals in aquaria can face several challenges despite the relatively controlled conditions. For example, declines in health state have been noted following months with high pollen counts. This review investigates the possibility of corals experiencing ‘hay fever’ and the possible pathways pollen could take which ultimately lead to reduced health and/or coral death. There are four pathways that may potentially cause such a decline in health. Additional nutrients may disrupt the delicate balance of beneficial microbes within the coral (causing dysbiosis), ultimately allowing potential pathogens to multiply and spread within the tissues. Pollen may also act as a carrier of pathogenic microbes or harbour contaminants such as iron, leading to the same or similar disruption and disease state within the coral. Pollen may also carry reactive oxygen species (ROS) to the coral. ROS have been shown to impact the oxygen concentration surrounding the coral, leading to oxidative stress, which again can result in cases of bleaching and/or disease.

**Abstract:**

Corals are facing a range of threats, including rises in sea surface temperature and ocean acidification. Some now argue that keeping corals ex situ (in aquaria), may be not only important but necessary to prevent local extinction, for example in the Florida Reef Tract. Such collections or are already becoming common place, especially in the Caribbean, and may act as an ark, preserving and growing rare or endangered species in years to come. However, corals housed in aquaria face their own unique set of threats. For example, hobbyists (who have housed corals for decades) have noticed seasonal mortality is commonplace, incidentally following months of peak pollen production. So, could corals suffer from hay fever? If so, what does the future hold? In short, the answer to the first question is simple, and it is no, corals cannot suffer from hay fever, primarily because corals lack an adaptive immune system, which is necessary for the diagnosis of such an allergy. However, the threat from pollen could still be real. In this review, we explore how such seasonal mortality could play out. We explore increases in reactive oxygen species, the role of additional nutrients and how the microbiome of the pollen may introduce disease or cause dysbiosis in the holobiont.

## 1. Introduction

Recent evidence suggests an aquatic origin to the influenza virus [1]. Specifically, viruses from the order Articulavirales, which includes influenza, have been shown to occur in Cnidaria (including corals). This novel family has been tentatively named the Cnidenomovirdae and argued to be the basal group within Articulavirales [1]. This would not be the first time that corals have been shown to harbour human-associated pathogens. For example, Herpes-like viruses are commonly found in coral tissues [2]. The bacteria *Serratia marcescens,* known to cause issues such as septicaemia, endocarditis and osteomyelitis in humans [3], is also known to cause white pox disease in the coral species *Acropora palmata* [4]. Even plant pathogens, such as those that cause tar spot disease in Sycamores (the fungus *Rhytisma acerinum*), have been shown to share the marine biome and cause a similar disease known as dark spot syndrome in corals [5]. Given this evidence of cross-over of pathogens between marine and terrestrial systems, is it possible that corals could be negatively affected by pollen? Pollen is certainly produced in vast amounts in many terrestrial systems but does it transfer to the marine biome? And what effect does it have on marine organisms? Here, we present an overview of potential mechanisms (summarized in Figure 1), by which pollen could have significant effects on coral health and mortality. We call for further research in this area, and end by making recommendations for the directions of such studies. Allergic reactions to pollen are experienced by humans on a global scale and ‘hay fever’ has been estimated to affect upwards of 500 million people per year [6,7]. It is known to be caused by hypersensitivity to specific allergens [8]. Although ‘allergens’ are broadly defined as any proteins or glycoproteins that are located on organisms or particles [9], hay fever is often induced by a range of organisms and particles including dust mites, moulds and pollen in particular [10]. Pollen is the most notable cause of seasonal allergic rhinitis, with symptoms (such as sneezing, rhinorrhoea and nasal congestion) occurring in humans exposed to pollen concentrations as low as 10 grains/m^3^ [11,12]. Here, we examine the potential for pollen to disperse into the marine system and its possible negative impacts on a coral host.

## 2. Pollen Production and the Perceived Problem

Seed plant reproduction depends on successful pollen production and dispersion. Some species rely on pollinators for dispersal whilst others utilise the wind. Although grains of pollen can vary in size from 7 to 617 μm, wind dispersed pollen tends to be both small and light as a result of being dehydrated to facilitate its dispersal [13].

Pollen levels vary seasonally; for example, in Europe, tree pollen levels peak from March–May, whilst grass pollen tends to occur from May–July, with flower pollen peaking during June–September [11]. As well as seasonal fluctuations, elevated atmospheric CO_2_ concentration can increase pollen production independently and in conjunction with the earlier arrival of spring [14]. Photosynthetic rates increase in plants exposed to elevated CO_2_ levels as a result of the additional carbon available [15]. Thus, with elevated CO_2_, flower abundance and size can increase, resulting in higher volumes of pollen being produced [16]. Caused by climate change, carbon levels are predicted to be as high as 730–1020 ppm by 2100 [17], and therefore the quantity of pollen production is likely to continue its upward trajectory. Consequently, timothy grass pollen production is estimated to Increase by approximately 200% by 2100 [16]. Such increases in pollen production will exacerbate the phenomenon of ‘nutrient cascades’, which are already occurring in forest ecosystems during peak pollen months [18]. The cascades of pollen provide large volumes of macronutrients to areas over short periods of time, a result which contributes to the observed high growth periods of forests in early summer [19]. However, a question arises. Do these cascades have more negative impacts for different organisms or ecosystems in general?

## 3. Linking Small Particle Dispersal with Coral

A small body of research has tried to explain the decline of coral reefs and, importantly, the lack of recovery of these reefs as well as the widespread geographic distribution of coral diseases by discussing the role of ‘dust’ [20,21,22,23]. Specifically, the hundreds of millions of tons of dust originating in Africa and Asia annually and transported at a global scale [21]. Dust plumes can form when areas of hot air develop [24] and are then transported by low pressure weather systems to a given destination [25]. Transportation can occur over vast distances, from the Sahara to Scandinavia for example [26]. Viable microorganisms, macro- and micronutrients, trace metals and an array of organic contaminants are also carried in dust air masses and deposited on both the land and in our oceans [21]. Following the addition of Saharan dust to water bodies, chlorophyll-a and bacterial activity within the water bodies has been shown to increase [27]. Further, bacteria have been shown to increase in abundance at locations that are often phosphate limited, such as coral reefs [28,29]. It is not unreasonable to hypothesise that a similar process is also occurring in ones home, in a coral reef aquarium—granted at a much smaller scale, but with the potential of as great disruption to the corals and the fish that call these places home.

## 4. How Would Hay Fever Manifest Itself in Coral?

One symptom of hay fever is rhinorrhoea, i.e., a runny nose [30]. Mucins are proteins in mucus that tangle to form a viscoelastic gel which is used to defend the organism from pathogens and other particles by acting as an impermeable barrier [31]. There are two main mechanisms that prevent particles diffusing through mucus. Particles either stick to the mucin fibres or are hindered by the small gap size between the fibres, which slows progression through the mucus [31]. Mucus viscosity is adjusted based on shearing rate, leading to the removal of the upper layer of the old mucus [32]. This can be influenced by a variety of mechanisms, e.g., swallowing [33]. In particular, viscosity tends to be lower at locations where mucus is produced [34] and hence the ‘runny nose’.

The function of mucus in corals is argued to be similar, i.e., simply put, a defensive measure [33,35,36]. It is theorised that the mucus in Cnidaria removes potential pathogens via methods such as sloughing [37], digestion [38] and acting as a medium for organisms (primarily bacteria) with antimicrobial properties [39]. Despite these known antimicrobial properties, some bacteria can use mucus as a growth medium [40] and, under stressful conditions, coral pathogens can become trapped within the mucus and thrive, inducing coral mortality [39]. Interestingly, bacterial growth is faster in mucus released into the surrounding water as opposed to mucus attached to the coral itself [40]. Theories relate this to the fluctuating oxygen availability and the presence of reactive oxygen species, or ROS [40]. Oxygen availability on mucus attached to the coral depends on whether photosynthesis or respiration is taking place, with oxygen-rich mucus occurring during daytime driven by photosynthesis and anoxic mucus produced during night time driven by respiration [40]. Additional ROS can inhibit bacterial growth and may be more abundant in periods of photosynthesis in coral-attached mucus [40]. Thus, the coral microbiome can change its function and community structure considerably over a period of 24 h [41], which is partly driven by the microenvironment and changing patterns of coral metabolism.

## 5. Fighting the Onset of Hay Fever

Not everyone’s immune system registers pollen as a foreign, harmful substance and as a result only some individuals experience hay fever [42]. Individuals experience hay fever due to hypersensitivity of the immunoglobin E (IgE) antibody. IgE is hypothesised to have developed to defend against toxins or parasites, which are becoming less prevalent in industrialised nations, whilst allergies appear to be rising [43]. Individuals may be exposed to varying levels or types of allergen and these come from a range of proteins with a variety of different functions [44].

Historically, hay fever and other allergic reactions have only been acknowledged to occur in vertebrates. This is due to the evolution of the adaptive immune system, which produces IgE antibodies [12,45]. IgE antibodies can only be formed in organisms that contain blood as they are created by the plasma cells found within the bone marrow and, importantly, only in mammals, in which it is theorised to have developed as the first line of defence against parasites [46]. However, pollen can also impact the innate immune system, i.e., the immune system found in both vertebrates and invertebrates, although this is largely understudied [47].

Correlational observations suggest that potential impacts on the innate immune system in corals deserve further consideration. Observations by the hobbyist community and aquarists associated with the trade of corals have noted a decline in coral health following months with peak pollen levels. This often manifests itself with tissue loss and can be devastating for a tank or system (Pers Obs Sweet and Thomas).

Innate immunity is a genetically determined immediate defence mechanism that appeared early in metazoan evolution. Its responses are generalised and designed to remove or limit infection of any pathogenic agent [37]; this can be done with pathways such as apoptosis, used in corals for both immediate immune responses and last-resort cell death responses to pathogens [48]. There are three common innate immune strategies in humans: detecting microbial ‘non-self’, detecting ‘missing self’ and detecting the common results of cell infection or injury [49]. In contrast, immunological memory is a key part of adaptive immunity [50], allowing the immune system to produce an enhanced response after coming into contact with a pathogen with which the organism has had a prior exposure event [51]. Adaptive immune systems evolved approximately 500 million years ago in jawed fish during the Cambrian era [52] in organisms that contained a pre-established innate immune system [49].

Conventionally, recombination activating gene (RAG)-mediated immunity is considered one of the standard mechanisms for adaptive immunity [53]. RAGs are utilised in combination with immunoglobin, T-cell receptors and the major histocompatibility complex to create the adaptive immune system in jawed vertebrates [54]. RAGs are also considered essential to the early stage development of T and B cells, which are a vital part of the adaptive immune system [55]. Cyclostomes, the other clade of vertebrates that contains only hagfish and lamprey, have convergently evolved their own version of the adaptive immune system [54]. Cyclostomes base their immune system on variable lymphocyte receptors, which are created by leucine rich repeats; the genes that form these are somatically diversified using gene conversion [54]. This convergent evolution has shown that despite the standard definition of adaptive immunity, it is possible for similar systems to develop using different mechanisms.

Further, despite being commonly utilised terms, many consider the strict definitions and separation of the innate and adaptive immune system to be out of date [50,53,56]. Indeed an overlap clearly exists between the two systems. For example, natural killer (NK) cells are known to form ‘learned’ responses to pathogen exposure, despite typically being considered part of the innate system [51]. As adaptive immunity evolved in the presence of innate immune systems, it is therefore likely to work in combination with the innate system as defence against threats, unlike the innate immune system which can respond alone.

The cellular immune response of corals has been explored in some detail. For example, the proposed coral ‘model’, *Stylophora pistillata* exhibits at least two cell types which show molecular markers indicating an immune function [57]. These cell types express genes typically associated with immune signalling pathways, antimicrobial activity and immune cell identity transcription factors [57]. Indeed, an adaptive-like immunity has been proposed to exist in scleractinian corals [37]. Corals have previously been found to have a self, non-self-recognition which was found as a result of allograft rejection in corals [37,58]. However, these concepts may also be outdated and the coral holobiont ‘concept’ may remove the need for an immune ‘self’ altogether [59]. As corals appear to lack an adaptive immune system, allograft rejection and other suspected immune responses may instead be triggered by a disruption to the holobiont [59].

So, to sum up, it is extremely unlikely that pollen allergens are a cause of any of the globally observed increases in mortality of corals (due to the lack of IgE in corals). However, there is substantial potential for pollen itself to ‘effect’ coral health, albeit largely in an indirect role.

## 6. The Potential Role of Pollen in Killing Corals

Pollen is comprised of a collection of compounds, most notably carbohydrates and proteins with the addition of amino acids, lipids, fatty acids, vitamins and minerals [60]. As the major constituents of pollen are carbohydrates and proteins, it is high in both carbon and nitrogen. Pollen grains also all contain an inner wall built with a pectin–cellulose combination and an outer wall, the exine, which is made of sporopollenin [18] and is covered in pollenkitt [61].

Many organisms harbour a stable or core microbiome, composed of the microorganisms consistently associated with an organism across space or time [62]. Pollen can also have a core microbiome. For example, 12 bacterial and 33 different fungal genera were found across eight different pollen species [63]. The likely high level of stochasticity involved in the formation of the microbiome of a pollen grain suggests that we should not expect strong consistency in pollen microbiomes at a global scale [64]. Indeed, pollen in urbanised areas is known to harbour relatively low microbial diversity [64] and pollen from wind pollinated species can have greater bacterial diversity in comparison to pollen from insect pollinated species [65].

With reference to potential links between pollen and corals, an important additional consideration is the ability of pollen to become contaminated with chemical elements, including strontium and iron [66], lead or aluminium [67]. These may have differing effects if corals are exposed to these contaminants. For example, strontium is a component of coral skeletons, and higher available levels can increase their growth rates [68], potentially making contaminated pollen beneficial. Conversely, corals exposed to more iron-enriched waters are more likely to experience mortality than those that are not, which is likely to be a result of iron inducing dysbiosis within the coral microbiome [69]. Exposure of corals to iron-contaminated pollen is likely to exacerbate this effect.

## 7. The Role of Reactive Oxygen Species and the Fight against Pathogens

Reactive oxygen species (ROS) are a normal by-product of both cellular respiration and photosynthesis, but despite this they are both reactive and toxic [70,71,72]. ROS build up can result in damaged proteins, lipids and nucleic acids [72]. Corals can experience a cellular stress response which occurs by producing an oxidative burst which has the potential to damage the coral–symbiont relationship [59,71]. Hyperoxic conditions, which more commonly occur during the day, are more likely to lead to ROS formation and oxidative stress [73]. Pollen may therefore act as a source of ROS in the system. Pollen also contains NADPH oxidases, which are used to help with germination [74], a source of ROS [75]. NADPH oxidase activity can trigger oxidative stress in humans even if the adaptive immune system has not responded [76], which suggests that this has the potential to also trigger oxidative stress in corals.

ROS build up can also occur in corals as a result of stress, resulting from, for example, increased UV radiation or thermal changes [71]. Corals can counter ROS damage provided they have a long enough recovery period without further stress. Indeed, both the coral host and their symbiotic algae have antioxidant mechanisms to reduce ROS build up and restore homeostasis [70].

Microbial associates, including several fungi and bacteria, have been shown to be beneficial to coral health [77]. Supporting the production of antibiotics [39] and secondary metabolites [78], as well as quorum sensing inhibition [79]. For example, *Pseudomonas*, one of the bacteria genera of the core pollen microbiome [63], has been found on corals suffering from disease and was theorised to be linked with recovery [80]. However, many others are thought to be opportunistic pathogens [81]. There are a plethora of reviews associated with the pathogenic nature of coral-associated bacteria (for example see [82]). In contrast, few studies have explored the role of fungi in coral health and disease [83]. As fungi are the dominant microbe present in pollen, it is worth considering the potential response of corals to these components of the microbiome. Chytrid fungi, such as *Batrachochytrium dendrobatidis*, a causal agent associated with the rapid global decline of amphibians [84], have been found to be involved in the release of nutrients from pollen into aquatic ecosystems [13]. Chytrid fungi have multiple life stages, from zoospores that can swim or ‘crawl’ to seek out specific nutrients such as proteins to a final cell with a fully developed rhizoid [85,86]. Rhizoid development in chytrids generates a feeding mechanism, having evolved to have a similar function to hyphae in other fungi. The rhizoid allows chytrids to use osmotrophy to feed [87]. Once they break through pollen walls they consume the pollen and then release their reproductive zoospores, which can be grazed by zooplankton and transferred up the trophic levels [13,86,88,89]. The fungi, should they be transferred from pollen to coral, may interact in a similar manner as with amphibians. In both cases, the first point of contact is likely to be the surface mucus layer, which is theorised to act as a nutrient source [90]. Should these fungi transfer to the coral, they may then limit the coral’s immune response by depleting the resources that the coral usually transfers into its associated mucus.

Fungi with hyphae, such as *Cladosporium* sp. [91], can similarly access nutrients inside the pollen grains after breaking down pollen walls or circumventing them via the pollen tube [18]. *Cladosporium* is one of the most abundant genera of fungi found within the core pollen microbiome [63]. This genus is also associated with lignin-cellulosic decomposition in marine ecosystems [92]. Such a process occurs as a result of their effective enzymatic systems [93]. The introduction of *Cladosporium* to aquaria systems with corals could therefore have a detrimental effect on the corals’ symbiotic algae, leading to bleaching or tissue loss.

Finally, fungi can also produce secondary metabolites which have been known to negatively affect humans [66]. For example, *Cladosporium* again is known to produce multiple secondary metabolites, including alkaloids and naphthalene derivatives. *Cladosporium* alkaloids could have both a positive and/or negative effect on coral as they have properties ranging from cytotoxic to antiviral [92]. Naphthalene derivatives similarly could be beneficial or harmful, with the derivatives having antimicrobial as well as antiprotozoal properties.

## 8. Pollen’s Alteration of the Coral’s Immediate Environment

In addition to introducing potential pathogens, pollen will almost certainly alter the immediate environment of a coral, i.e., the water column. Corals typically live in areas with low nutrient levels and rely on translocation to obtain the resources they need to grow [94]. In instances where sites have seen an enrichment in nutrients, there has been a corresponding increase in disease prevalence [95]. Indeed, long term enrichment of sites with nutrients may influence the spread of disease in multiple ways. For example, it may lower a coral’s pathogen resistance, increase microbe pathogenicity and/or alter the physiology of the coral host and symbionts, which would alter homeostasis with the coral holobiont [95].

The concentration of dissolved organic carbon (DOC) may also be increased by pollen in aquatic systems, and this could further destabilise the coral holobiont, leading to the increased growth rate of microbes living within the coral’s surface mucus layer [96]. The DOC could also disrupt the control methods that corals employ to maintain homeostasis such as limiting nutrient intake, which would allow an increased growth rate of microbes, potentially resulting in coral mortality via oxygen depletion, poison accumulation or microbial predation [96].

Pollen can also introduce nitrogen to coral systems. Nitrogen enrichment has been linked to coral bleaching and disease incidence [97]. Enrichment causes rapid *Symbiodiniaceae* proliferation, which disrupts nutrient transfer between them and their host [97,98,99]. Therefore, nitrogen limitation is an essential mechanism employed by corals to prevent being overpopulated by the algae [100].

## 9. How Do Corals React to the Presence of Pollen?

The simple answer to this question is that we do not know. However, we can speculate based on observed behaviour of corals in response to the presence of similar sized particles and how corals remove items identified as ‘non-self’. An example is the removal of microplastics. Microplastics are plastic particles under 5 mm, which includes nanoplastics which are suggested to be under 0.1 μm [101]. As pollen grains range between 7 and 617 μm [13], they fall firmly into the standard size range of microplastics. Microplastics are primarily removed from the water column by corals either passively (by adhesion) or actively (by ingestion). Passive removal occurs 40× more frequently than active removal [102], but corals can and do ingest particles of small sizes, i.e., between 0.2 to 1000 μm, with a ‘preferred’ size of under 400 μm [103,104].

Microplastics can act as a surface for microbial growth, reducing their buoyancy [105] and making them more available to be consumed by corals, and once ingested they can be found deep within the coral polyp [103]. Pollen grains can be similarly sized buoyant particles that contain their own microbiome. Should further microbial colonization take place, it may allow the pollen to sink enough to be available for coral consumption. Thus, pollen grains have the potential to act as a focus for the concentration of microbes from both distant (origins of the pollen followed by dispersal) and local environments, as well as a transport mechanism of these microbes into both the mucus layer and the coral body.

## 10. Future Research Directions

We clearly need to know more to determine whether pollen is the cause of this observed morality; therefore, future research is recommended. To determine how pollen reacts to coral, we should monitor both the environmental parameters and the direct response of the coral to pollen. Multiple species of coral would need to be used in future experiments to confirm that pollen affects coral health and does not influence just one species. Measuring the growth of corals via individual weights and 3D photometry would allow for growth rate monitoring, which would help to determine whether pollen limits coral growth. Both N-DOC and ICP-OES testing would be useful to monitor the effect that additional pollen has on water parameters. It would also be interesting to monitor the changes in microbial communities, particularly to see if pollen introduces any of its own microbiome to the coral. Monitoring for changes in mucus flow could also be an interesting way to determine whether pollen incudes an immune response. Further, due to the projected increase in pollen production linked with climate change, it would be beneficial to monitor how different volumes of pollen influence coral health. Should a negative response or correlation be found, monitoring of all these factors would provide key information that is necessary to help develop mitigation strategies to reduce any negative impact on corals in the future.

## 11. Conclusions

In summary, corals cannot and do not experience ‘hay fever’. Due fundamentally to the lack of IgE reactivity which underpins the physiological mechanism driving hay fever which, incidentally, is only seen in mammals. However, pollen could be having significant impacts on the physiology and health of corals by altering a coral’s microbial community which, in turn, is known to play an important role in coral metabolic processes and health. Alternatively, pollen might have more indirect effects such as driving up the levels of nutrients in the immediate environment. This could result in the induction of oxidative stress or potentially cause activation of the adaptive-like immune system in the coral. Therefore, there are many reasons why the study of pollen might be an important missing part of our understanding of coral ecology and global population and disease trends. This is exacerbated when we take into account the sheer amount of pollen produced globally each year, the fact this value is predicted to increase considerably under current climate change forecasts, the ability of pollen to travel long distances (upwards of 3000 km [106]) and pollens’ ability to concentrate microbes, fungi and chemical elements. When combining all this with the potential of pollen to act as a very effective delivery mechanism of these to corals, more research is certainly warranted.

## Figures and Tables

**Figure 1 biology-12-01469-f001:**
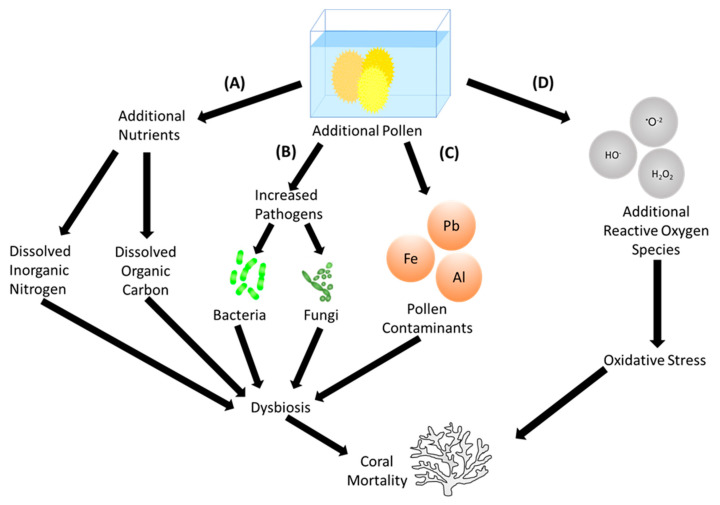
Potential routes to coral mortality upon the addition of pollen to aquaria. (**A**) Local nutrient concentrations could be raised by the breaking down of pollen grains, introducing substances such as dissolved inorganic nitrogen and dissolved organic carbon, leading to dysbiosis of the holobiont and eventually coral mortality. (**B**) Dysbiosis may be caused by additional pathogenic microbes carried by pollen into the system, leading to mortality. (**C**) Pollen may be a contaminated pollutant, whereby grains carry metals such as iron, aluminium or lead, which may then induce dysbiosis within the coral, leading to mortality. Alternatively, (**D**) pollen could cause coral mortality by carrying additional reactive oxygen species, causing corals to experience oxidative stress.

## Data Availability

Not applicable.

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
