# Peer review of "The Effect of Pollen on Coral Health"

_biology, 2023, doi:10.3390/biology12121469_

Round 1
Reviewer 1 Report
Comments and Suggestions for Authors
In this article, the author summarizes the impacts of pollen or particulate matter pollution on corals. Studying is interesting and holds significant importance. Overall, the paper provides ample evidence and effectively summarizes the existing literature in this field. However, there are certain questions that the author needs to address or answer.
1. In the abstract section, the potential impact of pollen on corals is mentioned but not elaborated upon. I believe it would be more appropriate to include this content in the introduction section.
2. There is a missing “.” before "A" in Line 61.
3. In Line 62, the figure mentions "Dissolved Inorganic Nitrogen" and "Dissolved Organic Carbon," but they are not explained in the figure caption.
4. In Line 66, the author extensively discusses pollen production and the influence of increased carbon dioxide on pollen abundance. I think this is unnecessary and can be shortened.
5. In Line 92, the author introduces the possible effects of dust on the coral ecosystem, but the subsection title is "Linking pollen dispersal with coral," which has not been discussed in that paragraph. It is suggested to modify the section title.
6. In Line 108, the author mentions that "runny nose" and mucus are the main symptoms of hay fever and explains the role of mucus in combating microbial invasion. Is mucus always present in corals, or is it only produced during hay fever? Can it serve as a marker for coral hay fever?
7. In Line 135, the author proposes that IgE in adaptive immunity is the main mediator of allergic reactions, but corals lack an adaptive immune system. Therefore, pollen-induced allergies are not the primary direct cause of coral death. Apart from IgE, do corals undergo processes such as microbial-induced cell apoptosis?
8. In Line 197, the author suggests three possible pathways for coral death caused by pollen: organic matter, pathogenic microorganisms, and heavy metal pollution, as an explanation for Figure 1. The author could include the part about heavy metal pollution in Figure 1 as well. Additionally, does the influence of organic matter in pollen occur as a byproduct of digestion or are these organic molecules already present on the surface of pollen? The author may need to provide a more detailed explanation of how the pollen wall's influence works.
9. In Line 223, the author lists pathogenic microorganisms that cause coral bleaching. However, do these microorganisms exist in pollen? And how are they transferred to pollen?
10. The author explains several possible pathways of pollen's impact on corals in this field. However, there is still a need for sufficient experimental evidence to support these claims. The author can provide insights on how to investigate the effects of pollen on corals from the perspectives of experimental design and analysis.
11. The article discusses the immune mechanisms of corals, and the author's team can incorporate open-source data on corals (such as Stylophora pistillata) to discuss coral immune cell responses.
Reviewer 2 Report
Comments and Suggestions for Authors
The manuscript by Barker et al. presents a review of the possibility of corals experiencing 'hay fever' and the potential impact of pollen on their health and survival. The authors conclude that corals cannot suffer from hay fever due to their lack of an adaptive immune system. The review is intriguing and may pave the way for future studies on coral microbiomes. There are a few minor comments, but the manuscript is acceptable.
-The author's conclusion is that corals cannot experience 'hay fever.' It would be preferable to avoid using questions in the manuscript's title.
-Line 36-59: In this paragraph, the example of viruses and fungi has been used to illustrate human-associated pathogens. I would recommend providing an example of a bacterial association between terrestrial animals and cnidarians.
-Line 47: "... is known to cause allergic reactions in millions of people." Please add references to support this statement.
-Lines 93-95: "A small body… 'dust'." Please add references to these statements.
-Lines 99-101: "Viable … our ocean." Please add references for this section.
